# Bedside functional monitoring of the dynamic brain connectivity in human neonates

Jerome Baranger ⬤ [1,6 ✉], Charlie Demene[1,6], Alice Frerot[2,3], Flora Faure[1], Catherine Delanoë[4], Hicham Serroune[1], Alexandre Houdouin[1], Jerome Mairesse[3], Valerie Biran[2,3], Olivier Baud[2,3,5,6 ✉] & Mickael Tanter[1,6 ✉]

Clinicians have long been interested in functional brain monitoring, as reversible functional losses often precedes observable irreversible structural insults. By characterizing neonatal functional cerebral networks, resting-state functional connectivity is envisioned to provide early markers of cognitive impairments. Here we present a pioneering bedside deep brain resting-state functional connectivity imaging at 250-µm resolution on human neonates using functional ultrasound. Signal correlations between cerebral regions unveil interhemispheric connectivity in very preterm newborns. Furthermore, fine-grain correlations between homologous pixels are consistent with white/grey matter organization. Finally, dynamic resting-state connectivity reveals a significant occurrence decrease of thalamo-cortical networks for very preterm neonates as compared to control term newborns. The same method also shows abnormal patterns in a congenital seizure disorder case compared with the control group. These results pave the way to infants' brain continuous monitoring and may enable the identification of abnormal brain development at the bedside.

[1] Physics for Medicine Paris, Inserm U1273, CNRS UMR 8063, ESPCI Paris, PSL University, Paris, France. [2] Assistance Publique-Hôpitaux de Paris, Neonatal intensive care unit, Robert Debré children's hospital, Paris, France. [3] Delegation Paris 7, Inserm U1141, University of Paris, Paris, France. [4] Assistance Publique Hôpitaux de Paris, Neurophysiology Unit, Robert Debré Children's hospital, Paris, France. [5] Division of Neonatology and Pediatric Intensive Care, Children's University Hospital of Geneva and University of Geneva, Geneva, Switzerland. [6] These authors contributed equally: Jerome Baranger, Charlie Demene, Olivier Baud, Mickael Tanter. ✉email: jerome.baranger@espci.fr; olivier.baud@hcuge.ch; mickael.tanter@espci.fr

Transition from fetal to postnatal life is a critical period exposing the developing brain to damage at a particular window of vulnerability. Any adverse events could have long-lasting and dramatic consequences on future neurocognitive outcome[1]. Consequently, continuous monitoring of blood pressure, heart rate, oxygen saturation, and body temperature are among basic routine care for any newborn requiring intensive care. Despite recent advance in newborn brain imaging, monitoring of cerebral function remains challenging in this vulnerable population. Functional magnetic resonance imaging (fMRI) led to major findings[2,3], demonstrating that MRI-assessed rs-FC (fcMRI) can identify alterations of functional cerebral connections in neonates[4]. fMRI suffers though from limitations including low accessibility, low portability, and high sensitivity to motion artifacts. Such limitations are critical when considering neonates and preterm infants. Other modalities such as electro-encephalogram (EEG)[5] or near infrared spectroscopy (NIRS)[6] can be used at the bedside but suffer from limited spatial resolution and cortex restricted assessment.

An interesting tradeoff between portability, sensitivity, and spatiotemporal resolution is offered by functional ultrasound imaging (fUS)[7]. It relies on ultrafast ultrasound at thousands of wave transmissions per second and its tremendous ability to capture microvascular blood flows (down to 1 mm/s in Ø250 μm vessels)[8]. This high sensitivity enables the detection of cerebral blood volume (CBV) variations, which are directly related to the underlying neuronal activity through neurovascular coupling[9]. Recently, fUS was successfully applied to neonates and revealed its potential to study sleep phases or epileptic seizures at the bedside[10]. As compared with other methods, fUS imaging allows serial, bedside imaging at low cost and without ionizing radiation exposure. It can be used by any operator familiar with ultrasound imaging.

Pursuing the quest of neonatal brain monitoring with fUS, we aim to propose a bedside monitoring technology able to identify markers of cerebral activity based on intrinsic functional connectivity (FC), potentially predictive of brain injury and neurological outcome. FC is defined here as temporal correlations between spatially distant changes in CBV related to neurophysiological events, frequently assessed during resting-state (rs). The value of rs-FC for studying neurodevelopment in newborns has been demonstrated using several technical modalities[11]. Moreover, rs-FC analysis already proved to be feasible with fUS in rodents[12]. The goal of this study is to investigate rs-FC on human preterm neonates using fUS for early brain function monitoring, and compare findings to term infants with normal and abnormal brain development.

## Results

### Static FC.
Based on our previous study demonstrating the feasibility of fUS imaging[10], we developed a miniaturized linear ultrasound probe anchored and stabilized in a custom 3D-printed headset to image the neonatal brain through the anterior fontanel (Fig. 1a). Brain parcellation is a prerequisite to compare cerebral activities of anatomical areas of interest. However, direct segmentation on 2D ultrasound images is quite challenging due to inter subjects' anatomical variability and inter-operator variability of the probe positioning. This issue has been resolved by collecting volumetric ultrasound data, which were obtained by mechanically steering the probe using a miniaturized rotational motor (Fig. 1b, Fig. S1 and Movie S1). The 3D ultrasound volume was used for the automatic registration of a MRI-based atlas in the ultrasound referential (Fig. S2 and Movie S2). We used the cerebral atlas of Makropoulos et al.[13], available for preterm infants from 28 weeks postmenstrual age (PMA), the average age

of the eligible population for this study. Figure 1c shows the results of this registration by representing the ultrasound-accessible volume over the full brain of a 28 weeks PMA neonate. The current configuration gives access to >40% of the brain volume (Fig. 1d and Movie S3). MRI atlas registration enables ultrasound navigation for choosing the fUS imaging plane of interest according to the cerebral structures to image and to assess.

We chose a coronal ultrasound plane to image symmetrical brain areas from both hemispheres parcellated according to MRI atlas (Fig. 2a) including the frontal lobes (FL) and the cingulate gyri (CG), the thalamus, caudate nucleus, subthalamic nucleus and lentiform nucleus. A specific advantage of fUS over other bedside compatible methods (EEG, fNIRS) is its ability to simultaneously image the cortical areas but also deeper subcortical structures. The CBV signal was averaged over all pixels within the atlas-defined areas and filtered in the <0.1 Hz frequency band. The rs-FC between two structures was obtained by computing the correlation coefficient between the two associated CBV time courses (Fig. 2b). This analysis was repeated in six preterm neonates born at $28 \pm 2$ weeks gestation and imaged at Day $9 \pm 3$ in several periods lasting 10-min each, during quiet sleep, as evidenced by EEG recording. It clearly appeared from the correlation matrices that the FL and the CG were highly connected in each hemisphere ($r = 0.8$ and $r = 0.9$, Fig. 2c, top left and bottom right of the correlation matrix). Moreover, high interhemispheric connectivity between the left and right FL and CG was evidenced by high coefficients in the matrix top right corner (Fig. 2c top right).

This interhemispheric intrinsic connectivity was also observed in the thalamus and to a lower extent in the caudate nucleus. Interestingly, these results are in good agreement with previous fMRI studies previously reported in preterm infants[14,15], but here, observed at the bedside in unrestrained newborns with a 10-fold increase in spatial resolution (respectively, 2.5 mm and 250 μm for fMRI and fUS). For comparison, the average static connectivity matrix obtained for four-term neonates is shown in Fig. S3.

### Pixel-mirror homotopic connectivity.
Then, in order to investigate the spatial resolution of fUS imaging, rs-FC was studied at the single-pixel level. Figure 3a shows a typical example of seed-based correlation (see Methods) obtained on a 5-min long window of quiet sleep recorded in one very preterm neonate. The two seeds S1 and S2 are placed in the left FL. Interestingly, the correlation maps reveal two different subnetworks, each presenting connectivity in the surrounding ipsilateral area and in its contralateral counterpart. These patterns suggest that the high spatial resolution of fUS rs-FC (<250 μm) gives access to fine-grained mapping of functional networks compared with those evidenced using rs-fMRI. Besides, it appears that mirror areas of the brain hemispheres, namely homotopic areas, have a high degree of connectivity. This homotopic connectivity can be assessed pixel-by-pixel as we assumed symmetric morphology between the ultrasound-imaged hemispheres (see Methods). Similarly to what is done in fMRI[16], pixel-mirror homotopic connectivity (PMHC) quantifies, for each pair of homotopic pixels, the CBV correlation coefficient computed and reported on a map (Fig. 3b and Movie S4). When FL, CG and thalamus rs-FC was analyzed, PMHC demonstrated a higher connectivity within the cortical layers of FL and CG corresponding to the gray matter (Fig. 3c). Conversely, thalamus did not exhibit this layered structure. These findings strongly suggest that PMHC is related to the underlying neural structure when studied at a single-pixel level. Furthermore, we demonstrated that PMHC was independent of vascular

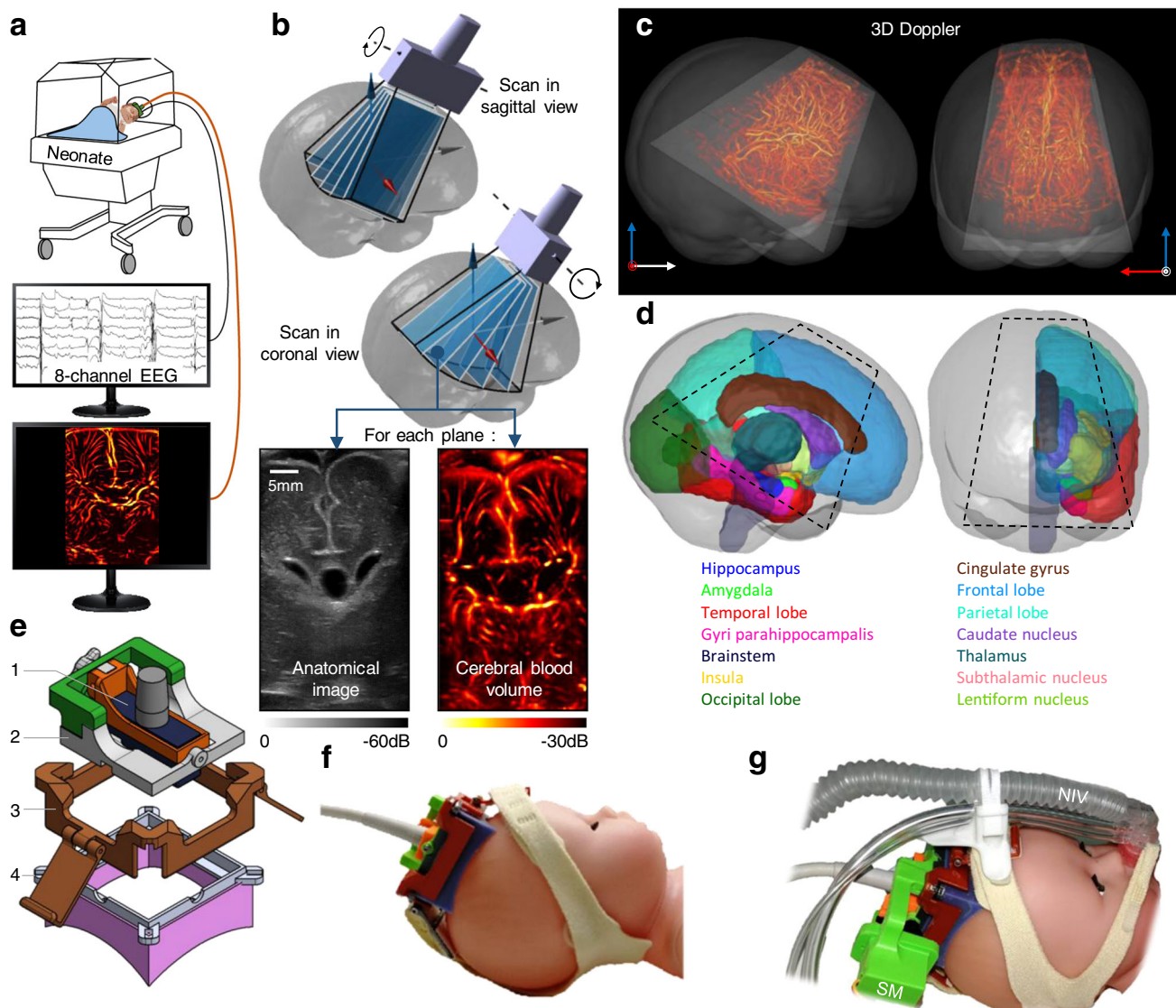

**Fig. 1 fUS recording and brain atlas registration. a** Simultaneous acquisition of an eight-electrodes EEG and fUS on neonates kept in their incubators. **b** 3D ultrasound tomography combining motorized plane-by-plane scans in coronal and sagittal views. The anatomic B-Mode image and the vascular doppler image are acquired in each plane. Only one scan in each orientation was needed for each patient. **c** In dark gray, sagittal and coronal view of a MRI T1 volume of a 28 weeks PMA preterm neonate. Trapezoid in light gray: brain volume accessible with fUS. In red, the 3D vasculature from Doppler tomography is overlaid. **d** Brain areas in the right hemisphere of a 28 weeks PMA preterm neonate, segmented according to Makropoulos atlas. Dashed black lines demarcate the ultrasound-accessible structures. **e** Detail of the 3D-printed headset. (1) Ultrasound probe. (2) Probe-holder with rotation axis. (3) Straps holder. (4) Cap with magnets to secure the probe-holder, and silicon casting adapted to the patient skull. **f** Ultrasound probe inserted in the headset, and secured to the model head using headgear's straps. **g** Motorized version of the probe-holder with as servomotor (SM), compatible with the non-invasive ventilation system (NIV) commonly used on very preterm patient.

density as shown in the Fig. 3d, e and Fig. S4, confirming the hypothesis of a neural origin for PMHC high values and excluding a vessel-size dependent vascular effect.

**Dynamic FC.** Beyond its high sensitivity and spatial resolution, fUS imaging also gives access to cerebral activity with a high temporal resolution. Hence, this technique enables to look at the "dynamic" functional connectivity (dFC) at a 1 s time scale instead of considering only "static" rs-FC on windows of several minutes. The dFC was assessed by comparing the instantaneous phase synchronization of each regions' time series (Fig. 4a, see methods)[17,18]. For two given CBV signals, their phase shift was assessed (via the cosine of their phase difference) every 1s, and

reported in the so-called phase matrix. As a proof-of-concept, only six areas were analyzed: the left and right FLs, CG, and thalamus, which had shown a high level of connectivity in the previous static rs-FC analysis. The rows and columns correspond to the different regions leading to a 6-by-6 dimensions phase matrix. The reference cohort included the six very preterm neonates previously introduced. They were compared with four term newborns imaged during the first postnatal week. In total, 10 independent acquisitions were collected for which all the resulting phase matrices were gathered in a unique data set (representing 50 minutes of acquisition, i.e., 3000 CBV images). In order to extract relevant information for this large dataset, we applied a data-driven clustering approach using the k-means algorithm (Fig. 4b) and derived from previous fMRI[17,19] and fUS[18] works.

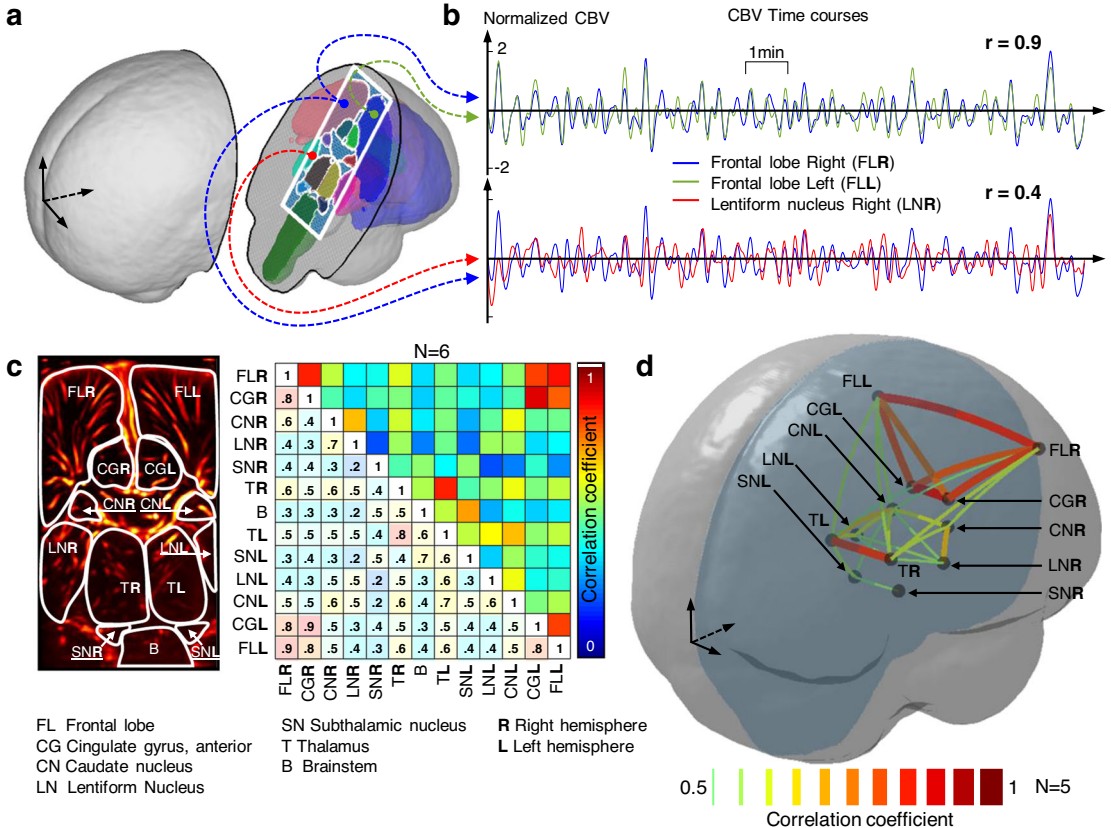

**Fig. 2 fUS reveals the intra- and interhemispheric connectivity in very preterm neonates.** (*CBV* cerebral blood volume). **a** Schematic view of the ultrasound plane of interest (white rectangle) registered on MRI atlas. **b** Example of normalized 10-min long CBV time courses obtained with fUS and averaged over three cerebral areas. The Pearson correlation coefficients *r* are computed between the signals. $r = 0.9$ for FLR vs FLL, $r = 0.4$ for FLR vs LNR. **c** Markropoulos atlas (white boundaries) overlaid on a doppler image in the plane of interest. The CBV is averaged in each zone and the correlation coefficients are computed for every pair of CBV time courses (see panel b for an example), leading to the adjacent connectivity matrix. This matrix is obtained by averaging the results over $N = 6$ very preterm infants born at $28 \pm 2$ weeks gestation and imaged at postnatal day $9 \pm 3$. **d** "Connectome" represented as links between different cerebral regions' centroids according to correlation coefficients. Source data are provided as a Source Data file.

The algorithm was tuned to return four cerebral connectivity states. Each of those four states is representative of a subgroup of close (in the sense of a L1 norm) phase matrices. In our data, state #1 represents the case where the six areas are in-phase, which means a synchronous behavior of CBV oscillations in these areas. This state can be interpreted as a thalamo-cortical connectivity network (Fig. 4b, right). State #2 shows in-phase behavior of FL and CG both inter and intra hemispheres. The thalamus also exhibits right/left in-phase signals. However, the thalamo-cortical dFC coefficients (FL or CG vs. T, left, and right) are weaker (0.33 ± 0.07). We interpret this as a decrease of connectivity between cortical and subcortical structures. State #3 shows the case where cortical areas and thalamus are out-of-phase, meaning that their CBV signal oscillations have opposite variations. Finally, in state #4, the CG left and right are connected to each other but appear to be in-phase-quadrature with FL and T. Conversely, FL and T left and right are in-phase.

Typical traces of phase matrix variations and state occurrence are shown in Fig. 4c and Movie S5. This example shows a term neonate spending most of the time in state #1, consistently with the previous static interhemispheric FC analysis. In contrast, the very preterm neonate depicted in Fig. 4c exhibits less occurrence of state #1 but more occurrence of state #2 and #3. This trend is confirmed by the statistical analysis in Fig. 4d, which compares the occurrence rate of each state for the full cohort of patient. The occurrence of state #1 (thalamo-cortical strong connection) for

the preterm group ($43 \pm 9\%$) as compared with the term neonate group ($68 \pm 10\%$) is significantly reduced ($p < 0.01$, $F = 16.9$, one-way ANOVA with post hoc Tukey's tests). Conversely, the occurrence of state #2 (thalamo-cortical disconnection) is significantly higher ($p < 0.01$, $F = 18.5$) for preterm ($29 \pm 6\%$) than for term neonates ($12 \pm 4\%$).

Even though our cohort size is too restrictive to draw clinical conclusions, we interpret this trend as a sign that prematurity can affect the development of thalamo-cortical connections. These results were consistent with similar findings previously obtained with fcMRI[20] or MRI tractography[21]. Interestingly, expanding the number of k-means states to five or seven does not significantly change the conclusions, as the same first four states are robustly exhibited (Fig. S5). dFC with fUS was consequently able to expose different dynamic networks between premature and term-born neonates.

Ultimately, we used the same dFC methodology to examine a rare clinical case that occured during the study time period: a term-born neonate with congenital seizure disorder showing a "burst-suppression" EEG state (see Methods). This patient was compared with the group of four healthy term newborns. In order to have enough phase-matrices representing the burst-suppression case (as compared with the term neonates dataset), we acquired three distinct data sets on this patient within the first 10 postnatal days (Fig. 5a). For this analysis, only four cerebral areas were retained (FLs and cingulate gyrus), as technical

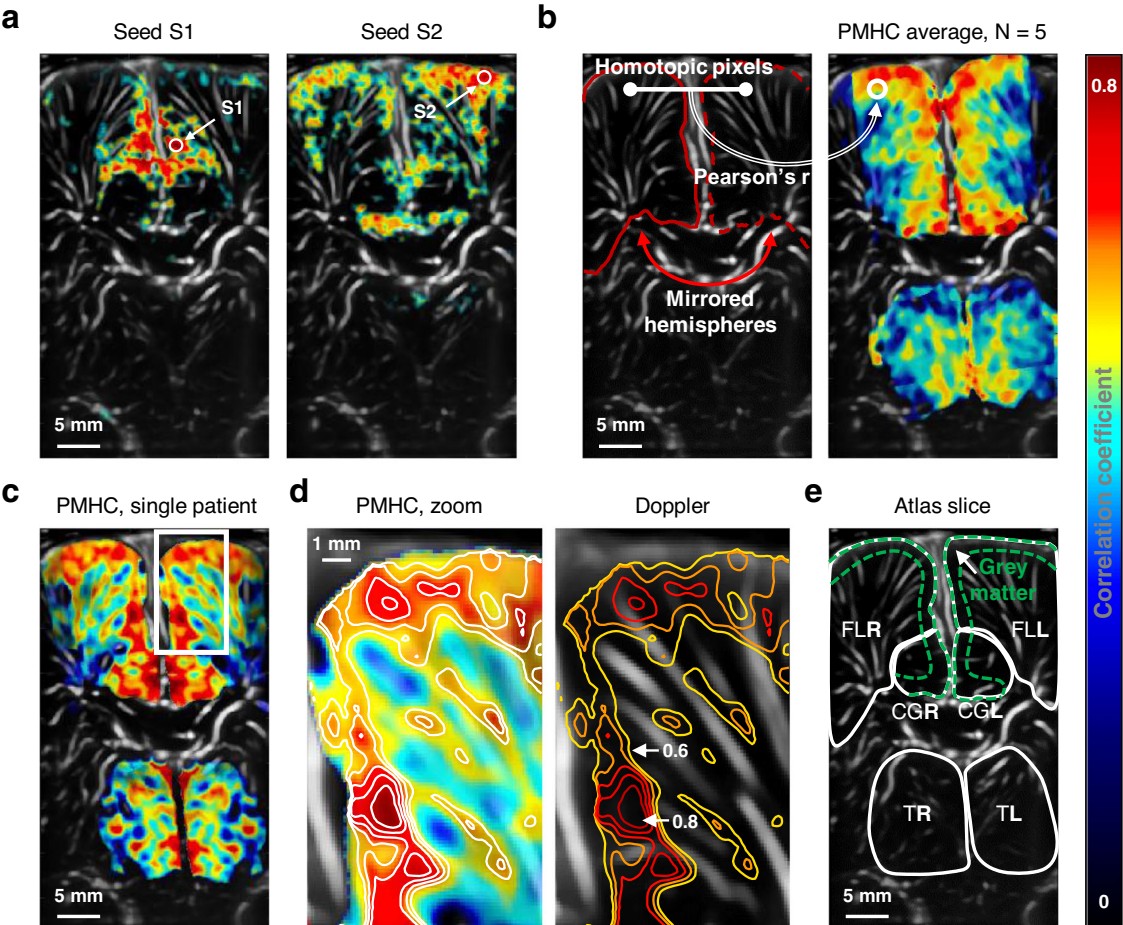

**Fig. 3 Seed-based correlation and homotopic connectivity leverage fUS high spatial resolution.** (*FL* frontal lobe, *CG* cingulate gyrus, *T* thalamus, *R/L* right/left) **a** Example of seed-based correlations from two seeds S1 and S2 in the left frontal lobe for a single patient. Thresholding on *Z* score ($p < 0.0001$, i.e, z > 3.72). **b** Pixel-Mirror Homotopic Connectivity (PMHC) map obtained by computing Pearson's correlation coefficients for every homotopic pair of pixels. Results are averaged for $N = 6$ very preterm neonates. **c** Example of a PMHC map for a single patient. A layered structure is visible in frontal lobe FL and CG. **d** Zoom on **c** white box. Several iso-lines are traced for *r* values ranging from 0.6 to 0.8, with a 0.05 step. Left panel shows the PMHC and right panel shows the corresponding vascular Doppler map. **e** Corresponding Markopoulos atlas slice showing gray matter segmentation in FL and CG.

limitations hindered the access to the thalamus for the patient with burst-suppression (see Methods). The clustering algorithm returned four states. State #1 shows total synchrony between FLs and cingulate gyrus, both inter- and intra-hemisphere. State #2 shows interhemispheric synchrony but disconnections between FL and CG. State #3 exhibits in-phase behavior in each hemisphere between FL and CG, but phase opposition between the two hemispheres. State #4 shows the left FL being out-of-phase with the three other areas, the latter being synchronous between each other. On the one hand, it appears that state #1 is preponderant for term neonates, with an occurrence rate of $80 \pm 7\%$ as compared with $45 \pm 16\%$ for the patient with burst-suppression (Fig. 5c). Conversely, state #3 and #4, which represents impaired interhemispheric connectivity are much more represented for the patient with burst-suppression, with occurrence rates of $20 \pm 12\%$ and $21 \pm 13\%$, respectively, to be compared with the occurrence of $0.02 \pm 0.01\%$ observed in both states in the term group. Even though several windows were considered for this single patient (allowing the report of error bars on Fig. 5c), these datasets cannot be considered as independent, therefore no statistical analysis was performed as $N = 1$ for the burst-suppression group. This "case study" analysis is presented to give a sense on how dFC could be used in a clinical framework with larger cohorts, to detect abnormal brain function.

## Discussion

This study shows the feasibility of rs-FC in the neonatal brain using fUS at the bedside, and its high spatial resolution as compared with existing techniques. rs-FC was assessed with three different paradigms: atlas-based static correlations, homotopic analysis, and phase-based dynamic connectivity, each one providing specific and complementary information. These results demonstrate that fUS imaging may provide a robust bedside imaging modality to monitor dynamically the emergence of functional networks in the early days of life and to quickly identify atypical developments and connectivity patterns. Moreover, its high signal-to-noise ratio (SNR), spatiotemporal resolution, and straightforward signal processing make it likely to be reproducible from one clinical center to another.

However, the current methodology has limitations, which should be tackled in future developments. Among these, the age difference inside the preterm group (at $28 \pm 2$ weeks PMA) has to be considered. Even though this gestational age group is commonly used in clinical studies to study very preterm patients, the brain rapidly grows during the early weeks of life. Thus, the cerebral areas morphologies evolve on a short time scale and this could hamper inter-patient averages. Nevertheless, the structures considered in this work have a limited shape variation within this group's age range, owing to a still limited gyrification in the plane of interest[13].

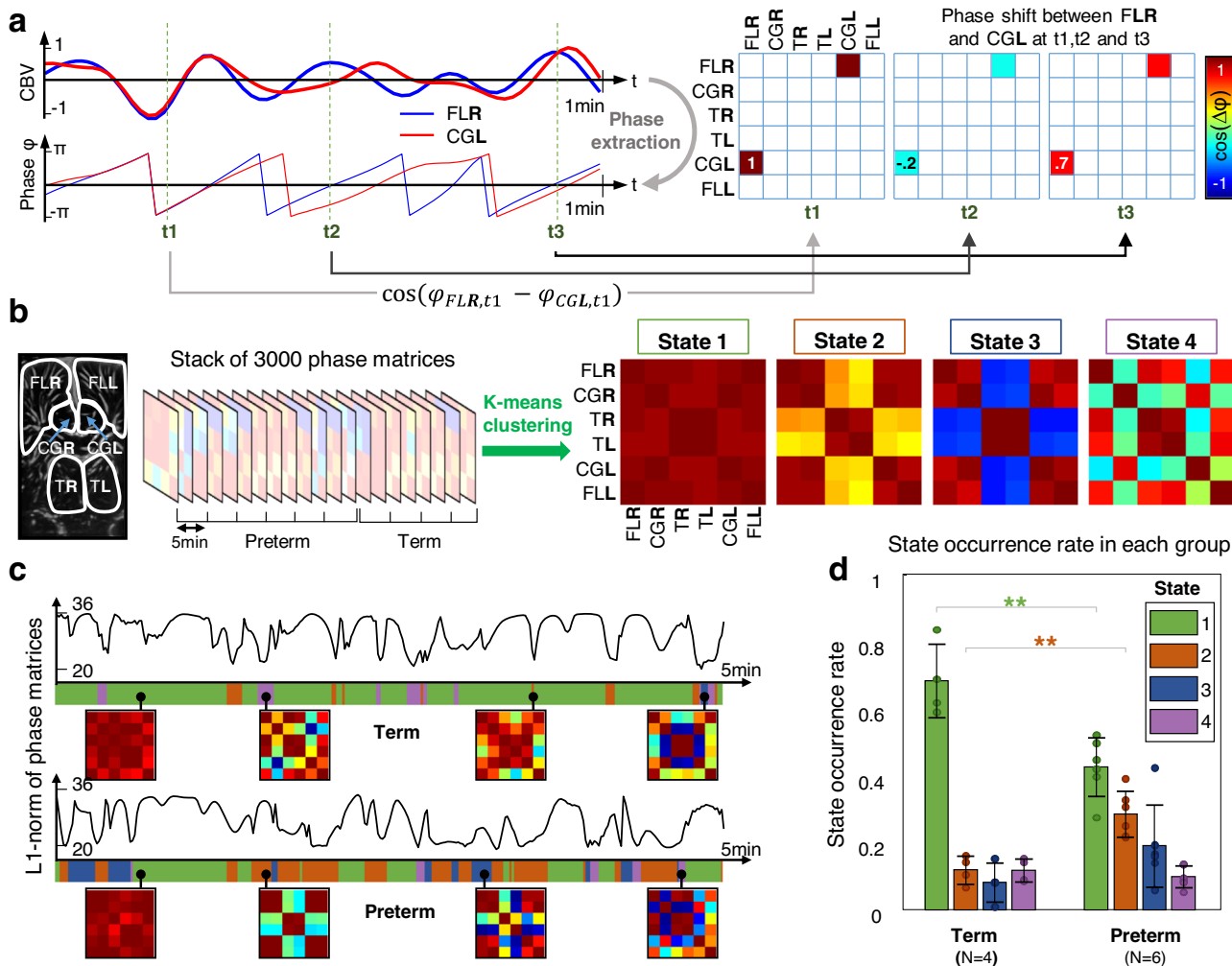

**Fig. 4 Dynamic connectivity reveals a disconnection of thalamo-cortical networks for preterm neonates.** (*CBV* cerebral blood volume, *FL* frontal lobe, *CG* cingulate gyrus, *T* thalamus, *R/L* right/left) **a** Construction of the phase matrices coefficients for the right frontal lobe (FLR) and the left cingulate gyrus (CGL). The instantaneous phase is extracted from the CBV time courses. The cosine of the phase shift between FLR and CGL is reported every 1s in the phase matrix. **b** Si cerebral areas are included in the analysis: the frontal lobes, cingulate gyri and thalamus in left and right hemispheres. The phase matrices are concatenated in the time dimension for 10 independent 5-min-long acquisitions involving six preterm and four term-born neonates. Unsupervised k-means with $L_1$ norm (i.e., Manhattan distance) is applied on the full phase matrix stack resulting in four connectivity states. The corresponding color scale is defined on the right of **a**. **c** Typical $L_1$ norm time course for acquisitions on one term infant and on preterm neonate. The fluctuations of these curves give information on the dynamic phase matrix changes. The closest state to the current phase matrix is encoded in the color ribbon below each curve. Four examples of phase matrices are shown in small square boxes for each patient. The colors in the ribbon representing the different states are defined in **d** legend box. **d** Occurrence rate of each state averaged on each population (**p = 0.003, one-way ANOVA with post hoc Tukey's tests). Error bars represent the standard deviation. Center of error bar is the mean value. N = number of independent patients in each group, with one acquisition per patient. Source data are provided as a Source Data file.

Brain growth was then considered as a mere dilation of the considered cerebral areas, allowing inter-patient comparison.

Besides, the number of labeled structures in the atlas and the MRI/US registration process is currently limited not only by the anterior fontanel window but also by the current MRI atlases available for both term and preterm infants. In the future, the additional use of the temporal fontanel window and more detailed MRI atlases at early stage of brain development may overcome these limitations.

The MRI-US registration process relies on segmented volumes such as cerebrospinal fluid (CSF)-filled structures. An inter-individual variability of the CSF-filled structures used for registration has been reported[22]. This process can be improved by adding anatomical landmarks including gyri, sulci, and great vessels, visible in both modalities (MRI and US). In the end, the relevance of the registration process relies mostly on the legitimacy of using segmented structures. Direct registration of 3D ultrasound B-Mode on MRI can also be considered but is beyond the scope of this study.

In addition, asymmetric morphology are common in neonates. If the patient exhibits strongly asymmetric hemispheres, homotopic connectivity computation can be compromised as the affine transform used to match the two cerebral hemispheres will not be accurate enough. fMRI studies on homotopic connectivity[16] also face this issue and address it by using more complex non-rigid registration tools such as B-splines based routines[23], which can also be applied to ultrasound.

This paper presents a first dFC analysis with fUS imaging on newborns, showcasing differences in thalamo-cortical network. Only six cerebral areas were used for this study. In order to capture finer dFC variations between tested population, increasing the number of areas should be considered.

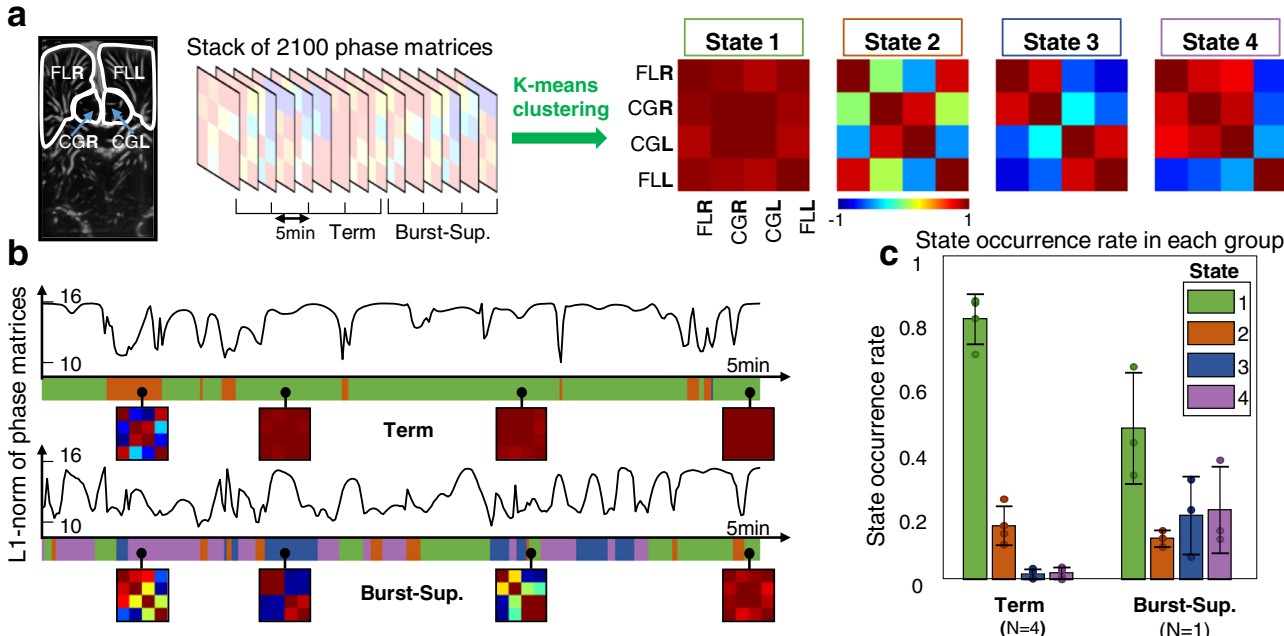

**Fig. 5 Dynamic connectivity reveals abnormal pattern in a newborn with burst suppression.** (*FL* frontal lobe, *CG* cingulate gyrus, *T* thalamus, R/L right/left) **a** Four cerebral areas are included in the analysis: the frontal lobes and cingulate gyri in left and right hemispheres. The phase matrix stack is the result of four independent 5-min-long acquisitions on four term neonates, and three acquisitions from the neonate with burst-suppression EEG pattern (Burst-Sup.). Four connectivity states are derived from the k-means algorithm on the phase matrix stack (see Fig. 4 for details). **b** Typical $L_1$ norm time course for acquisitions of one term infant and the neonate with burst suppression. The closest state to the current phase matrix is encoded in the color ribbon below each curve. Four examples of phase matrices are shown in small square boxes for each patient. The colors in the ribbon representing the different states are defined in **c** legend box. **c** Occurrence rate of each state averaged on each population (no statistical test because only one patient with Burst-Sup.). Error bars represent the standard deviation of either each patient assessment for term patients or each acquisition for the patient with Burst-Sup. Center of error bar is the mean value. N = number of independent patients in each group, with one acquisition per patient in the term group, and three acquisition per patient for the Burst-Sup group. Source data are provided as a Source Data file.

Regarding the fUS acquisition itself, the quality of the data are strongly related to the probe fixation headset. Most motion artifacts could be avoided with this innovative setup (see Methods). For future perspective, it should be noted that previous preclinical works have demonstrated the feasibility of fUS on freely moving animals[24]. The current setup also limits FC analysis to a 2D plane. This should be extended to 3D in the near future with the advent of matrix probes, already used in preclinical studies[25]. By then, the simple use of diverging beams with pediatric phased-array or micro-convex probes could give access to structures distant from the fontanel such as the temporal lobes. Deep gray matter structures (60–70 mm) can be easily accessed with the ultrasound frequency range used in this study (6MHz). Probes with smaller footprints could also be used to overcome the issue of narrow fontanels, especially for term-born neonates.

Regarding safety, attention should be given to acoustic power measurements[26]. The use of plane-wave instead of conventional focused beams strongly reduces tissue heating by ultrasound absorption. We used very conservative parameters for this study, with ISPTA values >20 times lower than the standards. Imaging through the fontanel with a small-footprint probe also reduced the risk of heating at brain-bone interface. Nevertheless, these safety considerations are crucial and the general ALARA principle ("as low as reasonably achievable") should always be followed. Long-term exposure to unfocused ultrasound has never been reported as potentially damaging for the tissues. However, constant attention and research on potential side effects of this method are still necessary, especially when studying vulnerable patients such as newborns.

Finally, we report here a proof-of-concept study based on small sample size but with low inter-individual variability. The added-value of this brain monitoring modality into routine care needs to be confirmed in larger clinical trials. Future works will typically involve a larger control group of healthy preterm neonates, compared to same-age patients with common brain injuries associated with prematurity.

Bedside imaging of dFC paves the way to a better assessment of the neonatal brain, similarly to recent fMRI studies[19], suggesting that human consciousness is strongly related to dFC patterns. In a more clinical perspective, many fMRI studies have identified alterations of rs-FC related to prematurity[27] or to autism spectrum disorder (ASD)[28]. Likewise, homotopic connectivity impairments were linked to ASD[29], even if the reproducibility of these fMRI observations is still a debate owing to the complexity and non-standardization of data denoising approaches[30]. We believe that fUS portability and practicality will bring new insights on these top priorities of public health.

## Methods

**Patient recruitment**. Nine healthy very preterm neonates (five males and four females) born at 28 ± 2 weeks PMA were included in the study. The data from the three first patients were used to set up the fUS headset design and therefore were not included in the quantitative study. Four healthy term-born neonates (two males and two females) in their first week of life were also included. Finally, one term newborn with a KCNQ2 mutation was imaged as well. This patient exhibited a pathological "burst-suppression" EEG pattern, associated with frequent ictal seizures. In general, mutations in KCNQ2 encoding the voltage-gated potassium channel Kv7.2 are found not only in 60–70% of families with the autosomal dominant self-limiting syndrome of benign familial neonatal seizures, but are also associated with a severe neonatal epileptic encephalopathy with early-onset epileptic encephalopathy within the first week of life, as reported here[31]. EEG at onset shows a "burst-suppression" pattern[5] in a majority of these severely affected patients. After the neonatal period, various patterns of seizures have been reported. Cognitive outcome varies widely from mild to moderate intellectual disability to severe developmental delay.

All the acquisitions were approved by the local ethical committee (Robert Debré Hospital, Comité de Protection des Personnes #120601, BELUGA protocol, promoted by INSERM, Institut National de la Santé et de la Recherche Médicale, French Health Institute). They strictly complied with the principles for medical research involving human subjects of the World Medical Association Declaration of Helsinki. Written and informed consent was obtained from all parents guardians.

**Ultrasound scanner and neonate headset.** The fUS acquisitions were performed at the patient's bedside by regular ultrasound operators. An ultrafast ultrasound research scanner (Aixplorer, SuperSonic Imagine, Aix-en-Provence, France) was used in conjunction with a custom linear ultrasonic probe (128 elements, 0.2 mm pitch, 6.4 MHz, Vermon, Tours, France) placed on the neonate's anterior fontanel. As fUS rely on the detection of subtle CBV variations typically during several minutes, it is therefore critical to ensure the stability of the probe on the fontanel. This constraint was addressed by conceiving a custom headset (Fig. 1a, e–g) combining a 3D-printed probe holder and a bio-compatible silicon casting adapted to newborns skull shape (Elite double 8, Zhermack, Badina Polesine, Italy). The headset was gently secured to the baby's head using a harness of straps. The silicon chamber was filled with ultrasonic gel and the probe could be positioned in a coronal or sagittal configuration. Then, it could be tilted manually or electronically using a small servomotor plugin (TowerPro MG91). The total setup weighed 50 g for the manual version, and 65 g for the motorized version. Installing the headset on the patient typically took <1 min. While imaging in conventional B-Mode, the position of the cap was manually adjusted on the fontanel to find the best acoustic window and to ensure the orientation of the acoustic plane. The headset was compatible with non-invasive ventilation system commonly used on very preterm neonates. As a 60 min clinical EEG recording was performed at the same time, the headset remained on the head during this period of time, even though fUS was only acquired during 20 min. This way, fUS acquisition did not interfere with the standard of care. No side-effect associated to fUS acquisition has been reported. The same headset was used for preterm and term neonates. However, as the morphologies of these two populations can be strongly different, the headset was designed to be adaptable to different skull shapes. Thus, for the same probe-holder, different caps can be designed to fit any kind of skull. This setup was used to maintain the probe in a fixed plan during fUS acquisition phase but also to acquire 3D ultrasound volume during the tomography phase (see Fig. 1f, g for details).

**EEG hardware and windowing.** In order to make sure that fUS observations arise from comparable neurological conditions, the wakefulness or sleep state of the patients as well as some physiological parameters (cardiac rhythm, respiration, and movements) were monitored. We used a Nihon-Kohden EEG-1200 system to record the EEG, electrocardiogram (ECG), and respiration. We used a standard eight-electrodes longitudinal bipolar montage in the 10/20 convention, placed around the ultrasound headset. The EEG device was synchronized to the ultrasound scanner using a TTL protocol. The preterm and healthy term newborns were observed during phases of quiet sleep. This sleep state was easily identifiable thanks to well-known EEG patterns such as the "trace discontinue"[5]. Quiet sleep was chosen because of patients' calmness, which limits the occurrence of motions artifacts. Typically, these phases last 20 min. As for the neonate with abnormal development, we focused on the burst-suppression phases due to their long duration (10–20 min) and to their clinical interest.

**Power doppler images.** Ultrasound can provide vascular information through the so-called Doppler modes. Particularly, the power doppler of a given pixel of the image has been known to be proportional to the fractional moving blood volume in this pixel. In the context of ultrafast ultrasound, the power doppler was acquired as follows. The tilted ultrasonic plane waves (−3°, 0°, and 3°), were successively emitted by the probe with a pulse repetition frequency of 1.8 kHz (PRF). Their backscattered echoes were coherently added and reconstructed into an image through a post-processing operation called beamforming. This resulting compounded image was therefore obtained at a frame rate of 600 Hz. A stack of 342 of such compounded images was acquired in 570 ms, followed by a pause of 430 ms enabling data processing and probe cooling. For this 1-second long elementary block, the tissue clutter was suppressed using an adaptive singular value decomposition (SVD) based filter[32,33]. This automatic SVD process ensure optimal and unbiased blood signal filtering. The remaining blood signal was integrated over the whole 342 frame stack, giving rise to a high-definition power doppler image. Such image was acquired in every scanned plane for the tomography phase or in a given plane every 1 s during 20 min for the fUS imaging phase. This sequence was far below the FDA requirements in terms of acoustic power, mechanical and thermal indexes (ISPTA = 30 mW/cm², MI = 0.5, TI = 0.5).

The SNR of the power doppler images is directly related to the number of plane-waves used for coherent compounding. Should the SNR be too low, it would be possible to increase the number of tilted plane waves and the PRF in each elementary block. It has to be noted that the PRF and the imaging depth were capped by hardware constraints. The greater the number of frames and the imaging depth, the higher the data amount to be processed for each block. The data pipeline has to be optimized to allow high PRF and deep images. This optimization work

was still in progress when the patient with burst-suppression showed up. This is why for this patient, the imaging depth was limited to 35 mm (vs 49 mm for other patients, after optimization). Recent scanners and computers have now overcome this limitation.

**Ultrasound tomography.** Using the optional motorized stage of the probe holder, a plane-by-plane scan could be automatically conducted in coronal or sagittal views. In each plane, both the ultrafast power doppler and the conventional anatomic B-Mode were acquired (Fig. S1). Knowing the 3D cylindrical coordinates of each plane, the plane-by-plane scan was interpolated on 3D cubic grid to form a volume. Once the motorized sagittal scan was over, the probe had to be manually rotated of 90 degree to perform the coronal scan. This operation was facilitated by the magnets (Q-05-1.5-01-N, Supermagnete, Gottmadingen, Germany) between the cap and the probe-holder. The cap remained on the patient's head, whereas the probe-holder was easily detached, rotated, and re-attached. As this manual rotation could cause slight cap displacements, the Doppler volumes obtained with the coronal and sagittal scans were automatically registered using a Matlab routine (*imregister*). The resulting geometric rigid transformation $T_{Tomo}^{Rigid}$ was applied to the B-Mode volumes as well. Doppler volume tubular structures were then enhanced with a Hessian-based filter[34] for better 3D rendering, namely, $V_{Doppler}^{Tomo}$.

Furthermore, the B-Mode images exhibited well visible hypoechoic structures filled with CSF, namely, the subarachnoid space, the lateral cerebral ventricles and the cavum septum pellucidum when it existed (Fig. S2). These structures can be easily segmented in any coronal and sagittal plane by many different image-processing techniques. We used the combination of two morphological reconstructions in Matlab (*imimposemin* and *imreconstruct*). This 2D segmentation allowed the 3D reconstruction of the CSF-filled structures $V_{CSF}^{US}$, which was used as a base for the MRI atlas registration step.

**MRI atlas registration and region extraction.** We use the "Consistent high-definition spatiotemporal neonatal brain atlas" from Makropoulos et al.[13], which was freely available from the brain Development project website (https://brain-development.org/brain-atlases/neonatal-brain-atlases/neonatal-brain-atlas-makropoulos/). This MRI-based neonatal atlas was chosen among several others[22] for three reasons: first and foremost, it was one of the few atlases available for very preterm infants born at 28 weeks PMA; second, it was established on a rather high number of participant at this age ($N = 5$); third, it came with a great number of parcels ($N = 87$). The CSF-filled structures were labeled in this atlas and gave access to the volume $V_{CSF}^{MRI}$. The two volumes to be registered $V_{CSF}^{US}$ and $V_{CSF}^{MRI}$ were first coarsely manually aligned to minimize the convergence time of the subsequent registration program. Then, we used a Matlab *imregister* routine to finely align the volumes and obtain the full geometric transformation $T_{US \rightarrow MRI}$. In order to compensate for small differences in shapes between the MRI atlas and the patient's anatomy, $T_{US \rightarrow MRI}$ was set as non-rigid (affine transformation comprising translation, scale, shear, and rotation). This allows a stretching of the different structures to improve the matching of the two volumes. This transformation could be also applied to the Doppler volume $V_{Doppler}^{Tomo}$ (Fig. 1c) in order to visualize the accessible cerebral volume. Finally, for any ultrasound plane, the transformation $T_{US \rightarrow MRI}$ gave access to the cerebral structures intersected by the ultrasound beam. Thus, these regions' boundaries were extracted and overlaid on the ultrasound plane to serve as a parcellation (Movie S1). Owing to inter-patient variability, the parcellation could be slightly manually adjusted to avoid aberrant segmentation: typically, this was to ensure that venous sinuses and cortical areas were not mixed up.

It should be noted that the registration performed here is applied to structures segmented on 3D ultrasound volumes, and to these same structures segmented in a MRI atlas. The inputs of the registration program are then binary volumes (voxels of one inside the structure of interest, and 0 outside), and not raw MRI or ultrasound data.

**Functional ultrasound signal pre-processing.** The fUS sequence is based on the repetition of the 1-second long power doppler elementary block described previously. As power doppler is proportional to the CBV, the fUS sequence gives then access to CBV variation along time. For each 1s-long block, only one cardiac cycle was retained to compute the power doppler image, in order to limit the potential influence of cardiac pace variation. After 20 min, when 1200 frames were obtained, a 2D rigid spatial registration was performed to align every frame with the first one. This was done to take into account the potential slight drifting of the cap. Besides, occasional patient brisk movements could cause cap motion on the skin and result in artifacts. The better the headset is secured to the head, the less frequently these artifacts occur. Therefore, it is technically possible to strongly reduce the occurrence of motions artifact simply by ensuring a tight headset setup. The remaining motion artifacts were automatically identified in the signal and the corresponding time points were rejected (Fig. S6). The CBV signal was then spatially smoothed with a Gaussian kernel whose full-width at half-maximum (FWHM) was 0.5 mm. The resulting signals were found to have dominant spontaneous fluctuations in the < 0.1Hz frequency band. This domain has been extensively described to be associated with resting-state FC dynamics[11]. Therefore,

the CBV was filtered in the [0.01 0.1]Hz frequency band using a bandpass fifth-order Butterworth zero-phase filter.

**Static connectivity: correlation matrices**. For each cerebral parcel in the ultrasound plane, the pre-processed CBV was averaged over all pixels. For the area #$i$, we note this temporal signal $CBV_i(t)$, with $\overline{CBV}_i$ being its temporal average and $\sigma_i$ its standard deviation. For a given number of parcels $N_{parc}$ and a given patient, the correlation matrix was computed over all the pairs of Pearson correlation coefficients as follow:

$$\forall (i,j) \in \left[\left[ 1, N_{parc} \right]\right]^2, r(i,j) = \frac{1}{n_t}\sum_{t=1}^{n_t} \frac{(CBV_i(t) - \overline{CBV}_i)(CBV_j(t) - \overline{CBV}_j)}{\sigma_i \sigma_j} \quad (1)$$

**Homotopic connectivity: seed-based correlation and homotopic transform**.
Seed-based correlation maps were computed by manually selecting a pixel and correlating its pre-processed CBV signal with all other pixels signals on 5-min-long windows. The resulting correlation map was thresholded by applying the Fisher's z-transform and choosing a level of significance of $z > 3.72$ ($P < 0.0001$, one-tailed test), corresponding to $r > 0.40$. The homotopic transformation $T_{R \to L}^{Homo}$, which links a pixel to its contralateral counterpart was obtained by automatically registering the mask of the right hemisphere $Msk_R$ on the mask of the left hemisphere $Msk_L$. These masks were obtained by simply fusing the parcels of the FL and the cingulate gyrus. As the midline was not perfectly centered, $Msk_R$ and $Msk_L$ were cropped to have a common surface. Then, $T_{R \to L}$ was fitted in Matlab by combining a reflection and an affine geometric transform. The PMHC was therefore obtained by computing the correlation for each pixel Pix as:

$$PHMC(Pix) = corr\left(CBV_{Pix}, CBV_{T_{R \to L}(Pix)}\right) \quad (2)$$

Where *corr* stands for the Pearson correlation. The resolution of $T_{R \to L}$ was measured by considering the mismatch between $Msk_L$ and $T_{R \to L}(Msk_R)$. This error was typically 0.5 mm, hence Gaussian spatial smoothing FWHM of the CBV was enlarged to 1 mm. Finally, in order to average the PMHC maps between patients, the different morphologies were aligned to a common reference. The first patient was arbitrarily chosen as the reference. The parcellation map of each patient was registered to the reference's parcellation using imregister routine with affine transforms. These transforms were applied to the corresponding PMHC maps, which were then averaged.

**Dynamic connectivity: phase matrices and clustering**. As correlation coefficients were only significant for sufficiently long period, we chose another estimator for assessing dFC: the instantaneous phase shift between the different regions' CBV time series. The implicit assumption was that the real signal $CBV_i(t)$ could be written as $CBV_i(t) = a_i(t)\cos(\varphi_i(t))$. Therefore, it is common to access the phase $\varphi_i(t)$ using the Hilbert transform $H_i(t)$ of $CBV_i(t)$:

$$\forall i \in \left[\left[ 1, N_{parc} \right]\right], \varphi_i(t) = \arg(CBV_i(t) + jH_i(t)) \quad (3)$$

with $j$ being the imaginary unit. At each time point, the phase shift cosine between two regions' time series gave a measure of their mutual synchronicity. Typically, a coefficient of one meant that the regions were in-phase, and 0 that they were out-of-phase. For two given parcel $p$ and $q$, we could then define the coefficient $m_t(p, q)$ of the so-called phase matrix $M_t$ at time point $t$ by:

$$m_t(p, q) = \cos(\varphi_p(t) - \varphi_q(t)) \quad (4)$$

Consequently, a fUS acquisition of $N_t$ time points and $N_{parc}$ parcels bound to a stack of phase matrices of size $[N_{parc}, N_{parc}, N_t]$.

A full stack of phase matrices was obtained over $N_{acq}$ independent fUS acquisitions by concatenating all the $M_t$ matrices in the third dimension, leading to a stack $M^{all}$ of size $[N_{parc}, N_{parc}, N_{acq} \times N_t]$. Here, we kept only $N_{parc} = 4$ parcels, with $N_t = 300$ seconds and $N_{acq} = 11$ windows (five acquisitions on preterm, three on term, and three on the pathological neonate). It boiled down to a stack of 3300 phases matrices.

Supervised clustering was performed on the complete $M^{all}$ stack using the Matlab implementations of the k-means algorithm (500 trials, random initialization, $L_1$-norm distance) with a parametric number of states $N_{states}$. The $L_1$–norm has been reported to be the most robust for this task (17, 18) and was kept for coherence with the literature. No significant changes in interpretation were observed when using $L_2$-norm though. For each state, the significance of the occurrence rates differences between the three populations was assessed by a one-way ANOVA followed by pairwise post hoc Tukey's range tests.

**Reporting summary**. Further information on research design is available in the Nature Research Reporting Summary linked to this article.

## Data availability
All data and software supporting the findings of this study are available from the corresponding authors upon reasonable request. Custom codes used for the collection of fUS data are protected by INSERM and can only be shared upon request, with the written agreement of INSERM. Source data are provided with this paper.

## Code availability
The code used to generate the results that are reported in this study is available from the corresponding authors upon reasonable request. Custom codes used for the analysis of fUS data used in this study are protected by INSERM and can only be shared upon request, with the written agreement of INSERM.

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

## Acknowledgements

We thank Sophie Pezet, PhD, and Olivier Villemain, MD, PhD, for their scientific advices. This research received funding from the Premup Foundation, the Chiesi Foundation Onlus and the European Union's Seventh Framework Program (FP7/2007-2013)/ERC Advanced Grant Agreement 339244-FUSIMAGINE.

## Author contributions

Contributor Roles Taxonomy format. Jerome Baranger: conceptualization, methodology, software, formal analysis, investigation, data curation, writing—original draft preparation, visualization. Charlie Demene: conceptualization, methodology, validation, formal analysis, writing—review and editing. Alice Frerot: conceptualization, investigation, resources, project administration. Flora Faure: data curation, formal analysis. Catherine Delanoë: Resources. Hicham Serroune: resources. Alexandre Houdouin: resources. Jerome Mairesse: investigation. Valerie Biran: conceptualization, validation, resources, supervision, project administration. Olivier Baud: conceptualization, investigation, resources, writing—review and editing, supervision, project administration, funding acquisition. Mickael Tanter: conceptualization, resources, writing—review and editing, supervision, project administration, funding acquisition.

## Competing interests

MT is co-founder and shareholder of Iconeus company which markets ultrasound neuroimaging scanners.
