## [Peer Review File · Nature Communications]

REVIEWER COMMENTS

Reviewer #1 (Remarks to the Author):

The presented study proof of concept paper in a small sample size of term and preterm infants. The authors demonstrate the feasibility to use ultrafast ultrasound to monitor brain function and brain connectivity in preterm and term infants at the bedside. Up to now functional imaging/ resting state functional imaging has been the domain of magnetic resonance imaging (MRI). The presented technique is novel and of high interest to all researchers and clinicians interested in the developing brain, perinatal injury and congenital cerebral abnormalities (e.g. obstetricians, neonatologists, paediatricans, paediatric neurologists). The ability to monitor brain function and connectivity at the bedside with a high time and spatial resolution would contribute to a better understanding of the pathophysiological bases of neonatal brain injury and congenital cerebral disorders. If this method will be developed further and implemented into clinical research and care it would potentially have a major impact on our current understanding of the developing brain and perinatal brain injury.

The manuscript is well written and concise, the statistical methods appropriate. The figures and movies are very helpful.

There are a few issues that I would like to address:

The sick term infant that the authors studied had a congenital seizure disorder (KnCQ2 encephalopathy) with a burst suppression EEG state. This is a very severe condition with a very severely abnormal brain function. It is therefore not very surprising that the authors find difference to healthy term infants and preterm infants. It remains to be shown if differences between healthy neonates and neonates with less severe brain abnormalities are detectable with this method. The authors should discuss this issue more explicit as it may not be clear to all readers how severe the condition of this sick term infants is.

The authors mention the high sensitivity of MRI to motion artefacts as a limitation of fMRI. However, it seems like the presented method is potentially also quite sensitive to motion artefacts. The authors should mention how robust this new method is with regards to motion artefacts and the explain the technical bases why this is different to fMRI.

Reviewer #2 (Remarks to the Author):

The authors propose a bedside system to carry out functional brain monitoring using ultrasound imaging on neonates. This is achieved by using the intensity of the received echoes as a surrogate for changes in blood volume, and provided that fluctuations in cerebral blood volume (CBV) are correlated with regional activation (proven in 10,12). The work builds upon previous work describing the technical implementation of functional assessment with ultrasound (10), and extends it to connectivity as already achieved in rodents (12). Connectivity is computed: 1) between brain regions, by measuring correlations between average CBV of different brain regions during deep sleep, 2) between homotopic areas, showing functional connection in the cortex area, and 3) between region of high resting-state connectivity, over time at high temporal resolution.

The proposed technique enables for the first time imaging of functional connectivity with ultrasound in neonates (this had been proven feasible in rodents by the same group in (12)). Effective translation of this technique into the clinic has tremendous potential for neonatal care. The technique has great potential to be exploited by other researchers, especially when 3D imaging is available (I don't see any reason why this should not be feasible).

The paper is well written and clear, and has most details to allow for reproducibility (but not all as detailed below). The reference list is adequate. The number of cases used in this study ($n = 5$) is relatively small; moreover, the neonate age variability of 4 weeks (28 ± 2 weeks) is large enough to create significant differences in brain structure and connectivity, as for example shown in (13) between 28 and 32 weeks. This may limit the validity of the inter-patient averages or at least make its interpretation difficult. Authors should comment on how they dealt with age difference, particularly differences in size and shape of the brain and its internal structures. Additionally, they derive statistical significance values from $N=8$ subjects (5 normal, 3 abnormal) and although they have a large amount of data for each, the data for each patient is most likely interdependent, hence although the results are promising, the statistical power is very limited to give any conclusive relation between the occurrence of states and the patient's profile. The results (and conclusions) from the experiment analysing the instantaneous phase synchronization should be revisited to either emphasize the limitations due to lack of sufficient data or including more data.

I think this paper is extremely interesting: it opens the door analysing functional connectivity in neonates in a rather simple, portable and translatable fashion which certainly is of interest to the broader community. The main limitation is the limited amount of patients included in the study which prevents drawing solid conclusions about the differences and similarities of functional connectivity information between patients.

The paper could be improved in the following aspects:

1. Data: including more patients would improve the statistical power and very especially for the third experiment on patient classification from high temporal resolution measurements.
2. Methodology: more specific details about the transducer and its 3D printed holder, and motorised system to collect 3D data, would be required to reproduce this research. It is not fully clear if the two orthogonal sweeps indicated in movie S1 are achieved by manually rotating the probe 90 degrees around its axis, or if both the mechanical sweep and the transducer rotation (from sagittal to coronal) are motorised and automated. Also, knowing how much manual user input is required in the pipeline, from image acquisition to obtaining the CBV maps: specifically, how much manual tuning is required for registering the 3DUS to the MRI, a notoriously challenging task that here is presented as solved, by first "coarsely" aligning the images, then using the Matlab routine `imregister`. These details are important to 1) understand how far we are from a clinical applicability, and 2) assess how generalisable the method is to other patients.
3. Figures: some figures (e.g. Fig. 4) are a bit difficult to read and should be improved (higher ppi should fix the problem).

Reviewer #3 (Remarks to the Author):

Short summary:

The study investigates resting state functional connectivity in preterm neonates compared to term neonates. Different techniques are used for this: atlas-based static correlations between intra and interhemispheric areas, static correlations pixel-by-pixel by seed-based correlation (pixel mirror homotopic connectivity) and phase-based dynamic connectivity. The intra- and interhemispheric connectivity in preterm neonates show that frontal lobe and cingulate gyrus are highly connected in the same hemisphere and in the contralateral hemisphere. Left and right thalamus are highly connected as well. The pixel mirror homotopic connectivity shows higher connectivity within the cortical layers of the frontal lobe and cingulate gyrus, independent of vascular density. The dynamic connectivity shows that both preterm and term neonates spend most of their resting state in state 1 and state 2. Where frontal lobes and cingulate gyrus are mostly in phase. In state 3 the left and right

frontal lobes and left and right cingulate gyrus are out of phase and in state 4 the frontal lobes are out of phase. One term neonate with burst suppression spends more time in state 3 and 4.

It was a pleasure to read this important research manuscript. I found the research original, of importance for future research, and very interesting as a potential clinical bedside biomarker of brain perfusion monitoring. The research seems very thorough.

Questions and remarks:

- The dynamic connectivity is shown for both term and preterm neonates. The static connectivity both pixel-by-pixel as the atlas-based connectivity is only shown for preterm neonates. What was the static connectivity for term neonates? Was there a difference between static connectivity in preterm and term neonates?
- I miss a more detailed interpretation of the different states described in the dynamic connectivity. Can the authors provide this?
- It is said that PMHC is independent of vascular density. Do the SVD-settings influence the number of vessels (aka the vascular density) that is visible?
- What was the SNR for the deeper structures, at the level of the thalamus? Could the SNR influence the results?
- What is the optimal depth for the probes used (in cm) and depth of the deep grey matter structures of the term infant?
- In order to assess the homotopic connectivity pixel-by-pixel, is perfectly symmetrical scanning required? Is perfect symmetric morphology possible? What happens if the anatomy of the patient is asymmetric, for instance, different size ventricles, or pathologies in one hemisphere? About 15-25% of extremely preterm infants have an IVH changing symmetry. Also, asymmetric ventricle sizes are very common in neonates.
- Was segmentation always possible and always successful?
- Patient demographics: any brain damage? Other pathologies?
- In fig 4d. Is significance testing even possible in such a small cohort (with n=1 for the burst suppression)?
- How long was the probe holder used in total per neonate, did it interfere with the care? How long does it take to install the probeholder in place and remove it? Any signs of use? Side-effects? Could the same probeholder be used for both term and preterm neonates? How heavy was it?
- What was the effect of fontanel size? Fontanel sizes are known to vary in size. Line 70. The authors state that more than 40% of the brain volume can be reached. However, to our experience, this partly depends on the fontanel size.
- Line 294 "as well as some physiological parameters (cardiac rhythm, respiration, movements) were controlled." ◊ controlled or monitored?
- Line 312 "followed by a pause of 430ms enabling data processing and probe cooling" – Was probe cooling necessary in order to achieve the TI of 0.5?
- Line 355 "The fUS sequence was made of the acquisition every 1 second of the Power Doppler image, whose signal was proportional to the CBV" this sentence might need rephrasing because it seems a bit unclear.
- In the materials and methods 20-minute periods of recording are described, are the 10-minute and 5-minute periods selected from the 20-minute periods? How?
- PHMC is used a couple of times instead of PMHC
- Fig S4 misses a colormap.
- Imregister is used before in Line 326, but introduced the second time Line 346
- Line 386 "PMHC maps were averaged between patients after a registration step aligning the different patient morphologies." ◊ How?
- The manuscript mentions little about possible safety concerns using relatively new technology. I feel that as this research may inspire groups around the world to follow this path, the authors should consider mentioning that there is still limited data for modern powerful diagnostic ultrasound equipment (Lalзад A, Wong F, Schneider M. Neonatal cranial ultrasound: are current safety guidelines

appropriate? *Ultrasound Med Biol* 2017;43: 553e60). There are also several animal models showing possible safety issues when long term scanning is applied. So, it might be wise to mention that safety issues should always be taken into account before clinical studies on vulnerable infants take place. Ongoing research into the bioeffects of ultrasound is, still necessary and risk versus benefit analysis should be performed by all operators (the ALAR principle).

It is possible to increase the number of infants in this study to include several common brain injuries seen as complications of preterm birth?

Minor comments:

- Line 37. The authors mention "low portability" when referring to (f)MRI. It might be better to mention that ultrasound allows serial, real-time, bedside imaging at low cost, without ionizing radiation exposure, and that it is nearly universally available.
- Line 49. The authors mention "ultrasound intrinsic functional connectivity". This sentence might need rephrasing.
- Line 286. Can the authors give some more information about the bio-compatible silicon casting? How was it made? is there a reference?
- The authors state no competing interests. Are authors involved within the company Supersonic?

Dr. Jeroen Dudink

Response to reviewers for the manuscript *Bedside functional monitoring of the dynamic brain connectivity in human neonates*

General introduction

We would like to thank the reviewers for their constructive comments, careful reading and evaluation of our work. We think that their insightful requests have allowed us to substantially improve the manuscript. We hope you'll find this revised version now suitable for publication in *Nature Communications*.

Sample size increase

We paid a special attention to the comments regarding statistical analysis and cohort size. We have understood that the dynamic functional connectivity (dFC) analysis that we showcased was impeded by the previous group analysis comparing a single neonate with “burst-suppression” with two groups of patients. It was confusing and inappropriate. Hence, we fully revised group comparisons. Between the two European Covid-19 waves, we were able to include one more preterm and one more term neonate in the study. The cohort size is now N=6 for preterm newborns and N=4 for normal term newborns.

New dynamic connectivity analysis: term vs. preterm neonates

We fully agree with reviewers' suggestion to compare first connectivity between full term and preterm infants, and we followed their advice. This analysis has definitely a higher clinical relevance. In the previous version of the paper, only 4 cerebral areas were used to conduct the dFC analysis : the frontal lobes and the cingulate gyri left/right. To better assess brain connectivity and detect differences between preterm and term neonates, we included another cerebral structure: the thalamus (left and right). This sub-cortical structure and its relationship with cortical areas were widely studied. The new Figure 4 presents the dFC analysis between preterm and term patients, with an increased number of 6 cerebral areas. Interestingly, it revealed a decrease of the occurrence of dFC states representing thalamo-cortical connectivity for preterm neonates as compared to the control term-born group (line 116-155).

Additional figure for the patient with burst-suppression

We believe that showing comparison between healthy full term infants (N=4) and a term neonate with burst-suppression EEG patterns remains of interest. We gathered the dFC analysis in a new Figure 5, of course without statistics due to small numbers (N=4 vs N=1). This figure can help to give a sense of the potential use of dFC in clinical framework, with larger cohorts and different pathologies. It has to be noted that for this dFC analysis, we used only 4 cerebral areas. Indeed, the thalamus was not visible for the patient with burst-suppression. Due to the very rare occurrence of such clinical cases, this patient had to be imaged while the ultrasound sequences were still under development, with an imaging depth capped to 30mm. Further technical developments overcame this limitation and allowed higher imaging depth. We reflected this consideration in the method section relative to *Power Doppler images* (line 409-413).

Reviewer #1 (Remarks to the Author):

The presented study proof of concept paper in a small sample size of term and preterm infants. The authors demonstrate the feasibility to use ultrafast ultrasound to monitor brain function and brain connectivity in preterm and term infants at the bedside. Up to now functional imaging/ resting state functional imaging has been the domain of magnetic resonance imaging (MRI). The presented technique is novel and of high interest to all researchers and clinicians interested in the developing brain, perinatal injury and congenital cerebral abnormalities (e.g. obstetricians, neonatologists, paediatricans, paediatric neurologists). The ability to monitor brain function and connectivity at the bedside with a high time and spatial resolution would contribute to a better understanding of the pathophysiological bases of neonatal brain injury and congenital cerebral disorders. If this method will be developed further and implemented into clinical research and care it would potentially have a major impact on our current understanding of the developing brain and perinatal brain injury.

The manuscript is well written and concise, the statistical methods appropriate. The figures and movies are very helpful.

There are a few issues that I would like to address:

1. The sick term infant that the authors studied had a congenital seizure disorder (KnCQ2 encephalopathy) with a burst suppression EEG state. This is a very severe condition with a very severely abnormal brain function. It is therefore not very surprising that the authors find difference to healthy term infants and preterm infants. It remains to be shown if differences between healthy neonates and neonates with less severe brain abnormalities are detectable with this method. The authors should discuss this issue more explicit as it may not be clear to all readers how severe the condition of this sick term infants is.

We thank the reviewer for this insightful comment and agree that previous comparisons of the analysis of dynamic functional connectivity (dFC) as it was initially presented was inappropriate. We invite the reviewer to refer to the introduction of this reply where a comprehensive response has been written regarding the important changes in data analysis. We hope that the new dFC analysis conducted between term and preterm neonates will answer to the reviewer concerns. We also added a more comprehensive description of the newborn with burst suppression in the method section (sub section *Patient recruitment*, line 341-348).

2. The authors mention the high sensitivity of MRI to motion artefacts as a limitation of fMRI. However, it seems like the presented method is potentially also quite sensitive to motion artefacts. The authors should mention how robust this new method is with regards to motion artefacts and the explain the technical bases why this is different to fMRI.

We thank the reviewer for his comment. We added some explanation regarding motion artifacts in section *Functional Ultrasound signal pre-processing* (line 462-466). With fMRI, the patient has to be immobilized into the imaging device, to limit the occurrence of motion artifact. With fUS, the imaging device is directly secured to the patient himself. Then, the occurrence of motion artifacts depends only

on the fixation system of the ultrasound probe on the neonate fontanelle. It has been demonstrated in preclinical works that fUS was feasible on freely moving animal, provided that a lightweight portable probe can be attached to their skull. Even though the physical constraint are obviously different for human neonates, the concept remains similar: motion artifacts can be mostly avoided if the probe is correctly secured to the patient's head. Additionally, the use of a spatiotemporal clutter filtering based on the singular value decomposition of ultrasonic data for the discrimination between blood flow and tissue motion cancels any remaining tissue motion artifacts due to tissue pulsatility or breathing.

Reviewer #2 (Remarks to the Author):

The authors propose a bedside system to carry out functional brain monitoring using ultrasound imaging on neonates. This is achieved by using the intensity of the received echoes as a surrogate for changes in blood volume, and provided that fluctuations in cerebral blood volume (CBV) are correlated with regional activation (proven in 10,12). The work builds upon previous work describing the technical implementation of functional assessment with ultrasound (10), and extends it to connectivity as already achieved in rodents (12). Connectivity is computed: 1) between brain regions, by measuring correlations between average CBV of different brain regions during deep sleep, 2) between homotopic areas, showing functional connection in the cortex area, and 3) between region of high resting-state connectivity, over time at high temporal resolution.

The proposed technique enables for the first time imaging of functional connectivity with ultrasound in neonates (this had been proven feasible in rodents by the same group in (12)). Effective translation of this technique into the clinic has tremendous potential for neonatal care. The technique has great potential to be exploited by other researchers, especially when 3D imaging is available (I don't see any reason why this should not be feasible).

The paper is well written and clear, and has most details to allow for reproducibility (but not all as detailed below). The reference list is adequate.

1. The number of cases used in this study ($n = 5$) is relatively small;
2. Moreover, the neonate age variability of 4 weeks (28 ± 2 weeks) is large enough to create significant differences in brain structure and connectivity, as for example shown in (13) between 28 and 32 weeks. This may limit the validity of the inter-patient averages or at least make its interpretation difficult. Authors should comment on how they dealt with age difference, particularly differences in size and shape of the brain and its internal structures.

We agree with the reviewer that the preterm neonate group exhibit age variability. Nonetheless, the 28 ± 2 weeks PMA range still correspond to the "very preterm" population. This group of very preterm patients is commonly considered in clinical studies as a relatively homogeneous population at high-risk of brain injury. We do agree that for future clinical studies, more restrictive age groups should be designed. For the present proof-of-concept study, we considered that changes in the cerebral areas assessed using fUS are quite limited between 28 and 32 weeks. Makropoulos *et al* (13) reported that

the subcortical areas (thalamus, sub-thalamic nucleus, etc) are mostly subjected to a volume expansion between 28 and 32 weeks. The gyrification process of the frontal lobes and cingulate gyri is also at its very beginning during this stage. Hence, age differences in brain growth appear to follow a homothetic transformation, allowing inter-patient comparison in the preterm group. More generally, the cerebral structure segmented in the atlas are large and coarse as compared to the resolution and size of the ultrasound images, which limits the impact of registration errors. We added these considerations to the discussion (line 188-194).

3. Additionally, they derive statistical significance values from N=8 subjects (5 normal, 3 abnormal) and although they have a large amount of data for each, the data for each patient is most likely interdependent, hence although the results are promising, the statistical power is very limited to give any conclusive relation between the occurrence of states and the patient's profile. The results (and conclusions) from the experiment analysing the instantaneous phase synchronization should be revisited to either emphasize the limitations due to lack of sufficient data or including more data.

We acknowledge that our first version of this manuscript had issues regarding the statistical testing, especially for the dynamic connectivity classification. We totally revised that part and included more patients in the groups. Because this comment was also formulated by other reviewers, we wrote a comprehensive statement about it in the introduction of this reply. No more interdependent acquisitions are considered for statistical testing.

I think this paper is extremely interesting: it opens the door analysing functional connectivity in neonates in a rather simple, portable and translatable fashion which certainly is of interest to the broader community. The main limitation is the limited amount of patients included in the study which prevents drawing solid conclusions about the differences and similarities of functional connectivity information between patients.

The paper could be improved in the following aspects:

4. Data: including more patients would improve the statistical power and very especially for the third experiment on patient classification from high temporal resolution measurements.

As mentioned above, we recruited more patients and revised our statistical analyses according to reviewer's comments.

5. Methodology: more specific details about the transducer and its 3D printed holder, and motorised system to collect 3D data, would be required to reproduce this research. It is not fully clear if the two orthogonal sweeps indicated in movie S1 are achieved by manually rotating the probe 90 degrees around its axis, or if both the mechanical sweep and the transducer rotation (from sagittal to coronal) are motorised and automated.

We added three more panels to the Figure 1 to provide more details on the ultrasound headset (different parts, closer look, position of the servomotor, compatibility with non-invasive ventilation). Additionally, we better explained how the probe had to be manually rotated of 90 degree around its axis

to achieve both coronal and sagittal scans (see section *Ultrasound tomography*, line 418-423). Using magnets and closely-fitted headsets and probe-holders, this manual rotation introduced only a small spatial shift, which was easily corrected with a rigid registration, as explained in the methods section.

Also, knowing how much manual user input is required in the pipeline, from image acquisition to obtaining the CBV maps: specifically, how much manual tuning is required for registering the 3DUS to the MRI, a notoriously challenging task that here is presented as solved, by first "coarsely" aligning the images, then using the Matlab routine *imregister*. These details are important to 1) understand how far we are from a clinical applicability, and 2) assess how generalisable the method is to other patients.

The registration part is indeed an important step of fUS methodology. Multi-modality registration is a wide field of research, not properly addressed in this study. We did not really achieve the direct registration of a raw MRI acquisition (T1 or T2w for instance) and raw 3DUS (B-Mode images), a challenge far beyond the scope of our work. Instead, we propose the registration of 3D structures segmented on 3DUS data and the same structures already segmented in a MRI atlas. This operation is much simpler as it deals only with binary volumes (voxels of 1 inside the structure of interest, and 0 outside) and can be performed with simple programs such as Matlab's *imregister* function. The "coarse" alignment step only refers to the initialization of the registration and is only used to minimize the convergence time of the algorithm. Considering the limited number of patient, we performed this operation manually as it only takes few seconds. It is though perfectly possible to automatize this step, with a low-resolution rigid registration.

The most critical part is the segmentation of the CSF-filled structures on 2D B-Modes images. The high contrast difference (60dB at least) between the surrounding tissue and these structures makes it easy to design an automatic segmentation program (here using Matlab's morphological reconstruction *imimposemin* and *imreconstruct*). From that point, the whole MRI-US registration process can be considered as fully automatic. The manually tuned parameters would then be the number of steps of the iterative registration algorithm and the increments size. Optionally, landmarks such as sulci or gyri can be manually added to constrain the registration. The positioning of these landmarks in the ultrasound images would involve manual tuning. To conclude, this process involves little to no manual tuning. Its relevance relies on the legitimacy (or not) of using segmented structures as input data for registration. We added further explanation on this topic in the *MRI atlas registration and region extraction* of the Methods section (line 443-444 and 452-455) as well as in the discussion section of the article main text (line 202-204).

Figures: some figures (e.g. Fig. 4) are a bit difficult to read and should be improved (higher ppi should fix the problem).

As requested, we increased the ppi from 220 to 330 and removed the compression of MS Word. HD figures in PDF format are also linked to the submission.

Reviewer #3 (Remarks to the Author):

Short summary:

The study investigates resting state functional connectivity in preterm neonates compared to term neonates. Different techniques are used for this: atlas-based static correlations between intra and interhemispheric areas, static correlations pixel-by-pixel by seed-based correlation (pixel mirror homotopic connectivity) and phase-based dynamic connectivity. The intra- and interhemispheric connectivity in preterm neonates show that frontal lobe and cingulate gyrus are highly connected in the same hemisphere and in the contralateral hemisphere. Left and right thalamus are highly connected as well. The pixel mirror homotopic connectivity shows higher connectivity within the cortical layers of the frontal lobe and cingulate gyrus, independent of vascular density. The dynamic connectivity shows that both preterm and term neonates spend most of their resting state in state 1 and state 2. Where frontal lobes and cingulate gyrus are mostly in phase. In state 3 the left and right frontal lobes and left and right cingulate gyrus are out of phase and in state 4 the frontal lobes are out of phase. One term neonate with burst suppression spends more time in state 3 and 4. It was a pleasure to read this important research manuscript. I found the research original, of importance for future research, and very interesting as a potential clinical bedside biomarker of brain perfusion monitoring. The research seems very thorough.

We deeply thank the reviewer for these positive and encouraging comments.

Questions and remarks:

1. The dynamic connectivity is shown for both term and preterm neonates. The static connectivity both pixel-by-pixel as the atlas-based connectivity is only shown for preterm neonates. What was the static connectivity for term neonates? Was there a difference between static connectivity in preterm and term neonates?

We added a new supplemental figure (Fig. S3) to present the results of static connectivity on term neonates and we added a reference to this new figure in the main text. In the caption of the figure, we explain that for term neonates, all the cerebral areas were not visible as compared to preterm neonates, due to the growth of the brain. Interestingly, the thalamo-cortical static connectivity is higher for term neonates.

2. I miss a more detailed interpretation of the different states described in the dynamic connectivity. Can the authors provide this?

As explained in introduction, we completely revised the dynamic connectivity section leading to a more appropriate statistical analysis. In the new text, we also provided a more detailed interpretation of the different states, as suggested (line 131-140).

3. It is said that PMHC is independent of vascular density. Do the SVD-settings influence the number of vessels (aka the vascular density) that is visible?

Indeed, SVD settings can influence the results, as the contrast-to-noise ratio will vary accordingly. To avoid introducing another parameter in the methodology (which would have to be tuned) we decided to use an automatic estimation of the SVD filtering thresholds. To do so, we used the results of a previous work from our group (Baranger et al, 2018, ref #28). This way, SVD filtering is adaptive and no bias is introduced by manual tuning. When we say that PMHC is independent of vascular density, we mean there is no correlation between high CBV values and high PMHC (see Fig. S4). We added a sentence in the *Power Doppler images* section (line 399-400) to highlight the importance of the adaptive SVD filtering.

4. What was the SNR for the deeper structures, at the level of the thalamus? Could the SNR influence the results?

SNR is difficult to measure on in-vivo data as it requires the segmentation of some vessels and some noisy areas. This segmentation can only be done on filtered data, in order to identify the vessels (not visible in B-Mode). The SNR estimation will therefore be intrinsically biased. Nevertheless, it can give a qualitative order of magnitude. We measured the signal in the thalamus of preterm patients, as deep as possible (~45mm). For noise, we took a region of interest within the ventricle. We found a SNR of 6dB. This has to be compared to the typical dynamic range of ultrafast Doppler acquisition, around 30 dB. It is likely that the SNR could influence the results, but considering the typical value obtained in deep structures, we had no reasons to believe that it did. As the ultrasound sequence used here is far below the FDA standards, the number of plane waves used for coherent compounding could be increased to improve the SNR. We added a sentence in the *Power Doppler images* section (line 405-407) regarding this topic.

5. What is the optimal depth for the probes used (in cm) and depth of the deep grey matter structures of the term infant?

The probe used had an elevation focus depth at 3.5 cm. For the study, we had an imaging depth of 49mm that allowed the visualization of most of the thalamus in term neonates. The same probe could be used with a deeper field of view. It already demonstrated its capabilities for images of 60mm depth in term neonates. Hence, we would expect the maximum depth of grey matter structures to be around 60-70mm. We added a statement about this in the discussion section (line 219-220).

6. In order to assess the homotopic connectivity pixel-by-pixel, is perfectly symmetrical scanning required? Is perfect symmetric morphology possible? What happens if the anatomy of the patient is asymmetric, for instance, different size ventricles, or pathologies in one hemisphere? About 15-25% of extremely preterm infants have an IVH changing symmetry. Also, asymmetric ventricle sizes are very common in neonates.

We thank the reviewer for this comment that highlights an important limitation of the methodology. This problem of asymmetry is indeed crucial for homotopic connectivity. In our study, the patients exhibited almost symmetric hemispheres, and no more complex registration routine was needed to find the homotopic transform. Nevertheless, this issue has been widely discussed in previous fMRI works. Most studies were performed with non-rigid registration using B-splines (rather than the mere affine

registration used here). This same routines are applicable to ultrasound data. We added a couple of sentence on this limitation in the discussion (line 205-209).

7. Was segmentation always possible and always successful?

Ventricle segmentation was always possible in our cohort as the contrast between tissue and CSF was always very high (60dB or more).

8. Patient demographics: any brain damage? Other pathologies?

No detectable brain damage or pathologies were reported for both preterm and healthy term patients, and the method section has been amended accordingly. The “burst-suppression” patient was the only one with brain abnormalities.

9. In fig 4d. Is significance testing even possible in such a small cohort (with n=1 for the burst suppression)?

Even though we included several acquisitions for the burst-suppression group (n=1, 3 acquisitions), we acknowledge that significance testing should be re-considered. This is why we entirely revised the dynamic connectivity section. The reviewer is invited to read the full answer to this concern in the introduction section of this reply.

10. How long was the probe holder used in total per neonate, did it interfere with the care? How long does it take to install the probeholder in place and remove it? Any signs of use? Side-effects? Could the same probeholder be used for both term and preterm neonates? How heavy was it?

We thank the reviewer for these questions, which are important for a clinical perspective. We added more details on the probe-holder in Fig. 1 and in the *Ultrasound scanner and neonate headset section*. The following text has been added (line 364-374):

The total setup weighed 50g for the manual version, and 65g for the motorized version. Installing the headset on the patient typically took less than 1min. While imaging in conventional B-Mode, the position of the cap was manually adjusted on the fontanelle to find the best acoustic window and to ensure the orientation of the acoustic plane. The headset was compatible with non-invasive ventilation system commonly used on very preterm neonates. As a 60min clinical EEG recording was performed at the same time, the headset remained on the head during this period of time, even though fUS was only acquired during 20min. This way, fUS acquisition did not interfere with the standard of care. No side-effect associated to fUS acquisition has been reported. The same headset was used for preterm and term neonates. However, as the morphologies of this two population can be strongly different, the headset was designed to be adaptable to different skull shapes. Thus, for the same probe-holder, different caps can be designed to fit any kind of skull.

11. What was the effect of fontanel size? Fontanel sizes are known to vary in size. Line 70. The authors state that more than 40% of the brain volume can be reached. However, to our experience, this partly depends on the fontanel size.

The reviewer is right, the fontanelle size is a key factor. Usually, the fontanelle tend to be large in very preterm patients. For full term neonates, a narrow fontanelle can decrease the signal, particularly in cortical areas. With time, we hope that the use of probes with small footprint (either pediatric phased-array probes or micro-convex probes) will be a way to overcome this issue. We added this comment in the discussion section (line 220-222).

12. Line 294 “as well as some physiological parameters (cardiac rhythm, respiration, movements) were controlled.” controlled or monitored?

We thank the reviewer for pointing out this mistake. These parameters were monitored, not controlled.

13. Line 312 “followed by a pause of 430ms enabling data processing and probe cooling” – Was probe cooling necessary in order to achieve the TI of 0.5?

No probe cooling was needed to achieve the TI of 0.5. The TI was measured relatively to the ultrasound sequence itself and did not take into account the pause between ultrasound blocks. It would have been the same with a shorter pause (or no pause at all). The cooling of the probe refers here to the probe surface temperature, not to the temperature rise potentially caused by ultrasound in the tissue.

14. Line 355 “The fUS sequence was made of the acquisition every 1 second of the Power Doppler image, whose signal was proportional to the CBV” this sentence might need rephrasing because it seems a bit unclear.

We thank the reviewer for reporting this sentence. We rephrased it (line 457-459).

15. In the materials and methods 20-minute periods of recording are described, are the 10-minute and 5-minute periods selected from the 20-minute periods? How?

Yes, the 10-min long and 5-min periods were extracted from the 20-min fUS sequence acquired for each patient. The rationale of choosing windows shorter than the maximum possible was to illustrate that connectivity results could be derived without long time averaging. Practically, these sub-windows are arbitrarily extracted from the 20-min acquisition, by taking the first 5-min long window with no movement artifact.

16. PHMC is used a couple of times instead of PMHC

We apology for this mistake which has been corrected.

17. Fig S4 misses a colormap.

This has been corrected.

18. Imregister is used before in Line 326, but introduced the second time Line 346

We thank the reviewer for noticing this. It has been corrected.

Line 386 “PMHC maps were averaged between patients after a registration step aligning the different patient morphologies.” How?

We thank the reviewer for his comment. We added the following statement in the *Homotopic connectivity: seed based correlation and homotopic transform* section (line 493-497):

Finally, in order to average the PMHC maps between patients, the different morphologies were aligned to a common reference. The first patient was arbitrarily chosen as the reference. The parcellation map of each patient was registered to the reference’s parcellation using imregister routine with affine transforms. These transforms were applied to the corresponding PMHC maps, which were then averaged.

19. The manuscript mentions little about possible safety concerns using relatively new technology. I feel that as this research may inspire groups around the world to follow this path, the authors should consider mentioning that there is still limited data for modern powerful diagnostic ultrasound equipment (Lalзад A, Wong F, Schneider M. Neonatal cranial ultrasound: are current safety guidelines appropriate? *Ultrasound Med Biol* 2017;43: 553e60). There are also several animal models showing possible safety issues when long term scanning is applied. So, it might be wise to mention that safety issues should always be taken into account before clinical studies on vulnerable infants take place. Ongoing research into the bioeffects of ultrasound is, still necessary and risk versus benefit analysis should be performed by all operators (the ALAR principle).

We thank the reviewer for this important remark regarding safety. We added the following statement to the discussion section (line 223-231):

Regarding safety, attention should be given to acoustic power measurements²⁶. The use of plane-wave instead of conventional focused beams strongly reduces tissue heating by ultrasound absorption. We used very conservative parameters for this study, with ISPTA values more than 20 times lower than the standards. Imaging through the fontanelle with a small-footprint probe also reduced the risk of heating at brain-bone interface. Nevertheless, these safety considerations are crucial and the general ALARA principle (“as low as reasonably achievable”) should always be followed. Long-term exposure to unfocused ultrasound have never been reported as potentially damaging for the tissues. However, constant attention and research on potential side effects of this method are still necessary, especially when studying vulnerable patients such as newborns.

20. It is possible to increase the number of infants in this study to include several common brain injuries seen as complications of preterm birth?

We were able to include one more preterm and one more term infant in the study between the two COVID-19 waves in France. However, we could not include more patients with brain injuries. The present proof-of-concept study paves the way for designing future larger clinical studies to address the relevant question raised by the reviewer. The effect of IVH, white matter damage or impact of intra-

uterine restriction growth on brain functions will be interesting targets for these future studies in larger patient cohort.

21. Minor comments:

- a. Line 37. The authors mention “low portability” when referring to (f)MRI. It might be better to mention that ultrasound allows serial, real-time, bedside imaging at low cost, without ionizing radiation exposure, and that it is nearly universally available.

We agree with the reviewer regarding these key features of ultrasound imaging. We are a little bit cautious when it comes to fUS as we do not want to overstate its possibilities. This method requires specific ultrasound scanners (with ultrafast-enabled hardware), and most commercially available scanners are not compatible. In this paper, we do bring a proof-of-concept of fUS feasibility and clinical in human neonates but real-time application of fUS is only envisioned. We added the following sentences to highlight the advantages of fUS (line 50-51): *As compared to other methods, fUS allows serial, bedside imaging at low cost and without ionizing radiation exposure. It can be used by any operator familiar with ultrasound imaging.*

- b. Line 49. The authors mention “ultrasound intrinsic functional connectivity”. This sentence might need rephrasing.

We removed the “ultrasound” from this sentence to make it clearer.

- c. Line 286. Can the authors give some more information about the bio-compatible silicon casting? How was it made? Is there a reference?

We added the reference of the silicon material (Elite double 8, Zhermack, Badina Polesine, Italy) in the method section (line 360-361). The casting was made using a 3D-printed mold, fitted to the headset 3D printed structure.

- d. The authors state no competing interests. Are authors involved within the company Supersonic?

Mickael Tanter co-founded Supersonic Imagine in 2005. He was shareholder and scientific advisor of the company and was also principal investigator of several research collaboration contracts between Supersonic Imagine and his academic institution between 2005 and 2014. Since 2015, he has no longer any links or interests with the company. Supersonic Imagine company was not involved in any part of this research study. This is the reason why the authors state no competing interests.

REVIEWERS' COMMENTS

Reviewer #1 (Remarks to the Author):

Thank you for the thorough revision of the manuscript. By adding additional patients to the data analysis the manuscript improved a lot. The authors replied to the queries of all reviewers sufficiently, I have no further queries.

Reviewer #2 (Remarks to the Author):

I thank the authors for the effort put to improve their paper based on the reviewer's comments. All the points I raised have been satisfactorily addressed and I am happy for the paper to be published in its current form.

Some minor remarks which could potentially further improve the paper, although should be taken as just suggestions, are the following:

1 - The number of cases is still relatively low. However I totally understand the challenge in acquiring the data, even more so in the current pandemic situation. I think delaying publication of the article to allow for more data would not really do any benefit to science, since the proposed technique seems very sound already and other researchers could benefit from it straight away. I do expect a clinically-oriented follow on paper with a larger cohort.

2 - In response to my remark on registration and variability of shape/size with age, authors state that "More generally, the cerebral structure segmented in the atlas are large and coarse as compared to the resolution and size of the ultrasound images, which limits the impact of registration errors". I understand the underlying sense of this sentence, but I would recommend insisting in the fact that non-rigid registration is also used to stretch the shapes to a given size and compensate for small differences in shape, which I don't think is clearly stated in the paper.

Reviewer #3 (Remarks to the Author):

Dear Authors,

I feel that all my comments and concerns have been well handled. I believe the manuscript has really improved by addressing all comments of the reviewers.

I enjoyed reading the new version. I have no further comments/suggestions.

Best wishes

Dr. J. Dudink

Response to reviewers for the manuscript *Bedside functional monitoring of the dynamic brain connectivity in human neonates*

Minor revision

General introduction

We deeply thank the reviewers for their positive feedbacks on the revised version of the manuscript. We believe that the paper has considerably improved thanks to their scientific critics and comments.

Reviewer #1

Thank you for the thorough revision of the manuscript. By adding additional patients to the data analysis the manuscript improved a lot. The authors replied to the queries of all reviewers sufficiently, I have no further queries.

We thank the reviewer for his appreciation of our work.

Reviewer #2

I thank the authors for the effort put to improve their paper based on the reviewer's comments. All the points I raised have been satisfactorily addressed and I am happy for the paper to be published in its current form.

We thank the reviewer for reviewing our work.

Some minor remarks which could potentially further improve the paper, although should be taken as just suggestions, are the following:

1 - The number of cases is still relatively low. However I totally understand the challenge in acquiring the data, even more so in the current pandemic situation. I think delaying publication of the article to allow for more data would not really do any benefit to science, since the proposed technique seems very sound already and other researchers could benefit from it straight away. I do expect a clinically-oriented follow on paper with a larger cohort.

We agree with the reviewer: this paper must be seen as a method-oriented introductory study. The methodology opens the door for clinically oriented studies that we expect to conduct in the coming years.

2 - In response to my remark on registration and variability of shape/size with age, authors state that "More generally, the cerebral structure segmented in the atlas are large and coarse as compared to the resolution and size of the ultrasound images, which limits the impact of registration errors". I understand the underlying sense of this sentence, but I would recommend insisting in the fact that non-rigid registration is also used to stretch the shapes to a given size and compensate for small differences in shape, which I don't think is clearly stated in the paper.

We thank the reviewer for pointing out this lack of clarity. We added the following sentences:

In order to compensate for small differences in shapes between the MRI atlas and the patient's anatomy, $T_{US \rightarrow MRI}$ was set as non-rigid (affine transformation comprising translation, scale, shear, and rotation). This allows a stretching of the different structures to improve the matching of the two volumes.

Reviewer #3

Dear Authors,

I feel that all my comments and concerns have been well handled. I believe the manuscript has really improved by addressing all comments of the reviewers.

I enjoyed reading the new version. I have no further comments/suggestions.

Best wishes

Dr. J. Dudink

We thank the reviewer for his positive appreciation.